# Staging as Communicative Activity of Shared Experiences a Way into a Fellowship for Deaf Children with Autism

**Janne Madsen * and Bjørk Kehlet**

Department of Educational Science, University of South-East Norway, 3918 Porsgrunn, Norway
* Correspondence: janne.madsen@usn.no

**Abstract:** We explore whether staging (a combination of symbols, drawings, paintings, and re-creating action using Playmobil figures) used as a communicative activity of shared experiences influences the sense of fellowship. This study is part of a combined research and development project, and the findings are canalised back into the daily work. The assistants are interviewed, and the data are analysed in a coding process. We found that (1) 'being confident' is important for belongingness. Staging supports a shared understanding and a shared feeling of belonging. We also found that (2) developing a shared repertoire of concepts is possible when staging, and this scaffolds the fellowship. However, the concepts and the communication must (3) be meaningful to all parties. One does not take part in communication without meaning, and when one does not participate, one is not part of the fellowship. Staging enabled Magne, the main person in this study, to express himself better. Moreover, when discussing a shared experience with the personal assistants, Magne seemed to communicate well and showed signs of belonging. We conclude that communication about shared activities and experiences leads to interest and engagement and fosters a shared feeling of 'doing something together' and being part of a fellowship.

**Keywords:** staging; drawing; re-creation of shared experience; fellowship; belongingness; AAC; autism

## 1. Introduction

Children with autism can experience difficulties with communication and relations [1]. Naturally, children with additional disabilities related to communication encounter even more comprehensive challenges and are at risk of loneliness. Magne, the main person in this study, is a 12-year-old deaf Norwegian boy with autism. He attends school in an alternative setting. His personal assistants carry out his schooling mentored by an educated teacher. The main goal of his schooling is to augment and develop his communication skills. The mode of communication is a combination of symbols, drawings, paintings, and moving Playmobil figures. We designate this process as staging [2], and we will describe it more thoroughly below. In this project, we explore whether staging as a communicative activity when re-creating a shared experience influences the sense of fellowship. Can staging be a way into a fellowship, including Magne and his personal assistants? Before presenting the context and the research methods, we describe the theoretical fundament.

## 2. Theory

Children with autism often experience difficulties related to the establishment of close relations. However, according to Umagami et al. [3], many autistic people are interested in social connections with other people despite sometimes experiencing difficulties with social interaction (p. 2118).

Dialogue as a fundament for development, communication, and relations in a fellowship, as presented in the cultural–historical theory, is our point of departure, and social interaction is considered highly important for development and learning [4]. Alternative Augmentative Communication (AAC), with its use of symbols, is organised systematically

according to themes and activities; it serves as our communication tool, in addition to photos, drawings, paintings, and the use of toy figures [2]. Language functions as the main tool in dialogues, as a cultural mediator, and as a tool for thinking [5]. An important argument for focusing on dialogue and language is the knowledge that young children obtain an understanding of their surroundings, relations, and other factors through conversations and interactions [6]. Conversations between adults and children in kindergartens often focus on events and activities, which are elaborated on and made comprehensible through language and actions. These kinds of conversations also influence the experience of belonging to a group.

### 2.1. Fellowship

Norway has approved the Salamanca Declaration, which states that all pupils should be part of a social and educational community, and no one should be excluded or segregated [7]. Pupils attending mainstream schools become part of a group or a class. By contrast, some special-needs pupils do not belong to classes. In Norway, some pupils (although this is a small group) are taught in an alternative setting and, thus, might be segregated from children their age. This is the context for the informant in this paper, as described in detail below.

According to Vygotsky [5], learning and development are two sides of the same coin. He described how children can master something that they could not achieve by themselves with support. With training and repetition, children gain individual mastery, and the assistance from experts can gradually fade out. This kind of learning is part of a collaborative process. Therefore, participating in a fellowship can support the sense of belonging to a group as it supports learning and development.

According to Wenger [8], a community of practice is characterised by mutual engagement, a shared repertoire, and joint activities. Mutual engagement is about interactions and relations among individuals in the community. Joint activities are processes where people negotiate and collaborate to reach shared goals. This negotiation is dependent on communication in dialogues. A shared repertoire utilises shared tools of communication to negotiate, but other tools such as shared routines and shared experiences can be part of this shared repertoire. We relate to Wenger's theory but use the concept of fellowship. To fully experience a fellowship, you have to belong to it. Wenger [8] described three different kinds of belongingness: engagement, imagination, and alignment. These concepts illustrate belonging as a state where one is restricted by group limits (alignment), and, at the same time, thinks beyond the limits and engages with the other members' feelings (imagination). The development of stories about shared practice can be a tool of belonging. According to Wenger [8], belonging to a community is not a locked position. One can work towards belonging through engagement, imagination and alignment. For most children, this kind of relational development will be a natural and fairly unconscious part of their development, but for a child with autism and without verbal language, one has to work more systematically when building a fellowship.

In 2022, Umagami et al. [3] published a review of autistic adults' feelings about loneliness. Loneliness seems to be a wide-ranging challenge for this group of people, and it would be relevant to try to prevent this using different approaches. Children and adults with autism desire friendship (p. 2129), but it seems difficult for them to establish these friendships. Social relationships where the participants share some interests and the person with autism experiences safety, recognition, and acceptance (p. 2129) can reduce feelings of loneliness.

### 2.2. Dialogic Teaching

Over several decades, researchers (see, e.g., Refs. [9,10]) have showed that learning occurs when pupils actively participate in dialogues, engaging with others, and making meaning. Dialogues are important for learning. In this project, our aim is to explore and develop dialogues for a child who is intellectually and communicationally disabled. In

this case, the traditional teachers' Initiative, pupils' Response, teachers' Feedback (IRF) structure for teaching is not suitable; given the subtle and unclear responses elicited from the pupil, it is difficult to assess whether the child understands [10,11].

Teaching communication is based on a dialogic understanding of the construction of meaning. Dysthe et al. [12] posited that *meaning is developed through dialogic interactions and collaborations between people situated in a context* (p. 46, our translation). Mercer, Littleton, and other colleagues explored dialogic teaching and exploratory talk as general concepts for pupils in early schoolyears and secondary schools and found that learning, understanding, and engagement increased [13,14]. Notably, there is a dearth of research on the dialogic teaching of mentally disabled pupils or those with autism.

According to Alexander [15], *dialogic interaction* describes a situation where both the teachers and the pupils ask questions, explain, and comment, leading to an exchange of ideas and the emergence of further questions. However, according to Boyd and Markarian [16], the form of the conversation is not the principal factor; reaching a level where both parties are actively engaged in the conversation is more important. Alexander [15] put forward five key principles that help us to identify dialogues: the conversation must be (1) collective, i.e., the participants address certain themes together; (2) reciprocal, i.e., the participants listen to each other and consider alternative viewpoints; and (3) supportive, i.e., the participants are confident enough to help each other reach common understandings. When conversating, teachers and pupils build on the ideas, so the dialogue becomes (4) cumulative. The last principle is that the dialogue should be (5) purposeful, i.e., steered towards specific goals (p. 3–4). Alexander's list of key principles can be used to assess whether communication can be defined as consisting of dialogues. This way, the list is helpful but also normative.

Muhonen et al. [14] found more children initiating dialogues in preschool than in Grade 1. They did not provide explanations in their research but referred to Smith et al. (2004) and Lehesvuori et al., Viiri, and Rasku-Puttonen (2011), who argued for more responsibilities for advanced age-level learning goals, limiting the time used for discussion, open-ended questions, and the exploratory approach to classroom dialogue (p. 151). Other researchers have discussed the differences that are 'inherited' in the culture of kindergartens and schools [16–18]. These differences might be even more substantial when instructing children with special needs because, according to Haug [17], special-needs teaching is more focused on subjects and tasks than ordinary teaching. This might indicate less space for dialogues, open-ended discussions, and exploratory approaches. In addition, Haug (p. 11) found that teaching pupils with special needs is often segregated from teaching ordinary classes, which also influences the possibilities of formal and informal conversations.

According to Gjems [6], the experience of being listened to and taken seriously makes children place themselves as active participants in dialogues (and learning). The dialogues should be about something meaningful to all the participants. These kinds of dialogues are basic elements when knowledge is constructed in a learning community [11,18] and when relationships are built between the participants.

The characteristics describing dialogue and communication indicate a border somewhere between the two concepts. With dialogue as the most restricted concept, expecting both parties to put forward questions and ideas and accumulate meaning, among others, the dialogue seems to be a more remote goal for children with communicative disadvantages. However, when one works with children's communicative competence, dialogue, as a situation where one explores, challenges, reconsiders, and extends ideas in ways that enhance learning [16] (p. 519), is currently developing as a parallel track. To ensure that one puts weight on children's responses, their understanding, and maybe even their experiences of meaning in communication, dialogue can be a supportive goal. According to Littleton and Mercer [19] and Rogoff [20], pupils who are guided to participate in meaning-making and independent thinking through dialogues can become learners who build on each other's ideas. This should be possible for almost all pupils in our schools,

although the basic ideas will vary between pupils with special needs and 'ordinary' pupils in a mainstream classroom.

According to Muhonen et al. [14], teachers play a significant role in creating opportunities for pupils' conceptual development and participation through inquiry, open questions, answers, and feedback, as well as in assisting pupils in explaining their thinking, seeking consensus, and solving problems together (p. 143). This kind of teaching is especially important for special-needs pupils and seems even more difficult with children who communicate less, in different ways, and by means of other communicative systems, such as AAC [21]. A way to support pupils in entering dialogues can be for the teacher to encourage, guide, and scaffold the pupils to express alternative perspectives and views [6,13]. Rogoff [20] used the term 'guided participation' and linked it explicitly to joint activities with a shared purpose and not the term 'scaffolding'. Even when teaching children with weak communicative responses, it is often still possible to guide them some distance towards increased participation.

In line with many researchers [5,6,10], we regard language as an important tool for learning to communicate. Muhonen [14] (p. 152) and Alexander [15] argued for the importance of providing pupils with opportunities to learn to ask questions, examine, and evaluate given ideas, negotiate solutions, and explain propositions. Again, these skills are premised on mastering a language and mastering basic communication, where one party expresses something while the other understands and responds. This is repeated in a shared engagement. All artefacts used in communication are considered language in our work. In this paper, we focus on staging by drawing, painting, AAC, and moving around Playmobil figures in the reconstruction and communication about shared experiences.

We now present the context, research methods, and discussion of the findings.

## 3. Context

Magne is a 12-year-old disabled boy who is deaf and autistic. He does not use verbal or sign language, so the development of shared language is built on AAC. His communicative responses are subtle, weak, and difficult to read and understand. Earlier, Magne's initial means of communication when he wished to do something was to simply to grab the hand of an adult and lead him to what he wanted. If this was unsuccessful, he would use inappropriate acting-out behaviours, such as hitting his head or biting or scratching the other person.

Children who can hear learn language by listening while simultaneously participating. Magne has never been able to obtain an understanding of his surroundings, relationships, and others through conversations and interactions because he cannot hear. Moreover, his autism has resulted in limited communication, which means that his general knowledge about his surroundings and communication is based on his personal experiences and is, therefore, very restricted.

Magne did not manifest much development in his understanding of communication between infancy and ten years of age when he started school at an alternative setting. His most developed method of self-expression was to grab the hand of an adult and pull them towards something he wanted, such as food, water, or a circular device to spin. This relatively simple form of communication was only meaningful in situations where the thing he wanted was visible. Magne could not express hunger or thirst. Expressing emotions was even more difficult for him. It is unclear whether he knows that hearing, non-autistic people communicate continuously. It is also unclear whether he experiences that kind of communication as meaningful.

Furthermore, Magne acts out inappropriately when he does not know what he is expected to do or he experiences numerous changes during the day (e.g., contact with many different people: teachers, taxi drivers, personal assistants, parents, siblings, and their friends, etc.). Because of the acting-out episodes and the need to use restraint in these situations, two personal assistants are always on duty to look after him during the day [22].

### 3.1. Teaching Related to Daily Activities at Home

Magne started school in a class where pupils with special needs were taught in a separate location within the school. Two years ago, Magne switched to schooling at home, which constitutes schooling in an alternative setting. Magne was segregated from his class for two main reasons: (1) his acting-out behaviours are unpleasant and, at worst, dangerous for the other pupils in the classroom, and (2) Magne becomes stressed and frustrated when he encounters too many different people during the day. Therefore, Magne does not belong to an ordinary fellowship of children. The personal assistants in his schooling at home might compensate for the lack of fellowship with classmates to some degree.

Two personal assistants implemented the teaching plans made by an educated teacher. The teacher is paid by the local community, and teaching is conducted in accordance with the public curriculum but adapted to Magne's severe special needs. Plans and reports are given to the local education authority. Magne's education revolves around communication related to the routines of daily life (ADL) and practical activities. School, understood as systematic, continuous teaching, and learning communication, begins when Magne is woken up in the morning, and continues until his bedtime.

The main goal for his schooling, in addition to ADL, is to develop his communication and interaction skills. We are open to different forms of communication. According to Boyd et al. [16] (p. 517), the form of communication follows function, not the other way around, and we are more concerned with the functionality of the communication between Magne and his personal assistants than the form. We use boards with Widgit symbols organised according to themes [21] (Figure 1). For each theme a board is made. Meal with bread (Norwegian: brødmat) is the theme for the board shown in Figure 1. On each board you see symbols and each symbol is combined with the written word, to make sure the assistants use the same symbol each time a concept is used in the conversation. An example is the lower left corner where the symbol with the two vertical lines means "to take a break" (Norwegian: pause), and the symbol just above means "to wait" (Norwegian: vente). When the assistants use the boards, they explain, ask, tell, and answer by pointing to the symbols. They also expect Magne to point to the symbols once he understands their meanings. Symbols that are related to concrete, visible things (nouns) and things that Magne recognises as useful or important, have proved easier for him to learn.

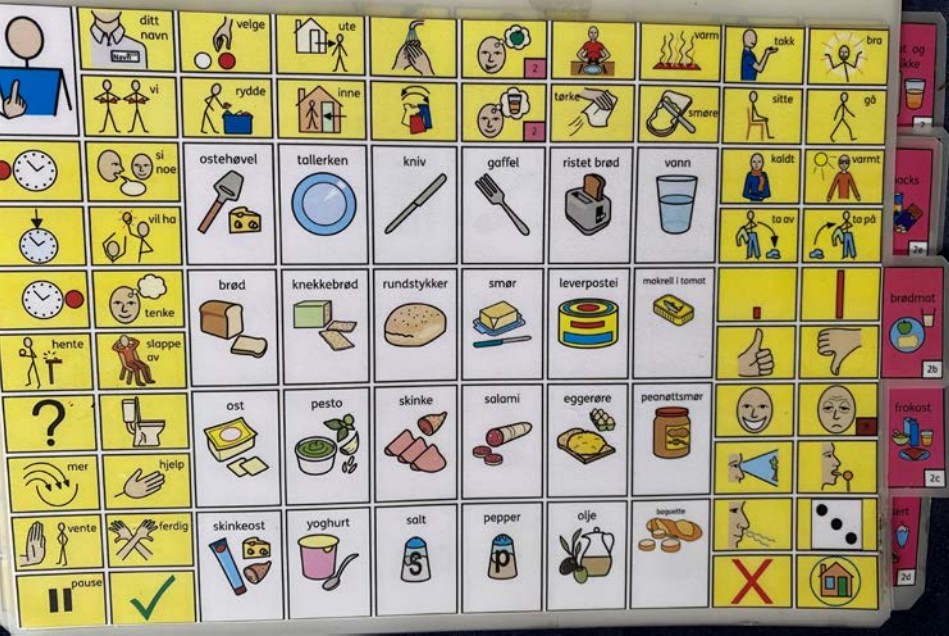

**Figure 1.** Photo of a page from the book with AAC symbols and text (in Norwegian) [21].

The educational tasks are connected to practical activities such as splitting wood, baking bread, and others to communicate with Magne about (1) what happens now, (2) what has happened, and (3) to plan future practical activities. In this paper, we mainly explore communication focusing on what has happened. The question 'What did we do together?' and the experience of shared activity and belonging to the fellowship are discussed. Magne has no adequate language with which to communicate, and both he and the personal assistants have to learn to participate in their shared dialogues. It is not easy to know what Magne understands. For the reasons stated—deafness, autism, and intellectual disability—he does not respond like other children his age. To know about Magne's experiences and what he finds meaningful, we put weight on communication with Magne as an active participant. According to Wells [13], we introduce an inquiry model, where both the personal assistants and Magne explore and 'discuss' different issues that make meaning not only for the adults but also for Magne. The process of learning this form of communication is not only a matter of importance for Magne but also for the staff. This process is organised as action learning, which we will now describe further.

In our project, shared activities are practised each day but not based on fellowship as a goal. One of these activities has been chosen at random to analyse for this paper. In this activity, Magne and the two personal assistants, Anne and Bendik, go for a walk in the forest, pass some small becks, slip on some ice, throw stones into the water, experience rainy, freezing weather and wet shoes, and so on. The activity is documented with photographs. One assistant communicates with Magne along the road, but it is difficult to use AAC, pointing to symbols on a board while walking. Magne answers now and then when they stop.

Back home, a sheet of paper about 1.5 m × 0.5 m in size is taped to the floor. The parking space, trees, becks, inlet, ice on the track, and others seen on the walk are drawn on the paper while the assistant continuously 'talks' to Magne about the drawings using AAC symbols, photos, and body language. Photos are glued onto the sheet in the relevant places, and a Playmobil® car of the same type and colour as 'his' car is parked in the parking space. Three Playmobil people, with photos of Magne's and the two assistants' faces glued onto them, walk from the car and into the forest, cross the becks, pass the ice, throw stones into the inlet, and walk back to the Playmobil car before driving back along the drawn road to Magne's home (represented by the AAC symbol for 'home'). Through this use of staging, the shared experience is retold using as many valuable communicative possibilities as possible.

### 3.2. Action Learning: To Learn and Develop Simultaneously

In the daily work with Magne, we, the staff, the teacher, the parents, and the researchers, learn from our experiences, actions, and the theory at the same time as Magne learns. Action learning has been our basic long-term strategy. The teacher who is responsible for the school, the management, and the group of personal assistants implementing the daily plans have all agreed to emphasise communication as an important task for the schooling. The personal assistants are very engaged in this task.

The experiences and awareness of very subtle elements of communication as germinal for dialogues and as a basis for discussions and reflections between the assistants enhance the shared awareness and seem to increase Magne's engagement. According to Elden and Levin [23], increased engagement, enhanced collaboration, and shared awareness are not unusual in the processes of action learning. To experience these with a severely disabled child further stimulated our work, and we proceeded to phase 2, building on our action learning in phase 1. We described the implementation of phase 1 in another paper [24] with the stages (1) asking the questions and defining the challenges, (2) planning the action, (3) taking action and (4) evaluating the action, before carrying on to the next phase with similar stages [25]. In phase 2, we expanded from action learning to a combination of action learning and action research, through the stages. We undertook collaborative research with the aim of developing a degree of shared understanding and associating this understanding

with the consequent actions [26]. We spent a relatively long time in the first phase and simultaneously developed our understanding of the use of AAC and communication with Magne. This is what Wertsch [27] described as shared understanding.

We now describe our research method.

## 4. Materials and Methods: Data Gathering and Analysis

The methodology used in this research project is qualitative. Based on observations and phenomenological interviews with only a few informants, generalisation is not possible. However, the situation is thoroughly described above, and the empirical findings are discussed in relation to the theoretical fundament. We hope that future research texts based on our findings will promote and inspire development in other contexts and function as a secondary cultural artefact [28].

### 4.1. Informants

The informants in this research are Magne and his assistants. Magne cannot answer questions as an informant, but he is observed by persons knowing him well. The assistants and the parents determine whether participating in a research project is best for Magne; if not, the project will be stopped. Informed consent is obtained from all of them based on information about the project [29].

Six personal assistants work in the dayshift with Magne: three with a bachelor's degree in educational science and three with a lower education level. They work closely together in long shifts and learn from their experiences and one another. Altogether, they are a very knowledgeable group. They work in teams of one educated and one less educated tandem. Both genders are represented almost equally. They have been working with Magne for approximately four months to two years. One team (i.e., two of the assistants) is randomly chosen for the interviews.

Both researchers are skilled workers. One functions as the daily leader who is responsible for leisure activities, while the other is responsible for the school plans. None of us works as a personal assistant. We are also relatives of Magne. The inputs of the staff and their experiences are essential in regulating the interpretations of Magne. The research findings are discussed critically in staff meetings as inputs to the further process, as described above. The findings are also critically reviewed by academic colleagues as a counterweight to any potential subjectivity arising from our closeness to Magne. To augment credibility, the action spirals described above, and the method described below are followed to the letter.

### 4.2. Data Gathering

Data were gathered from the entire group of six personal assistants working the day shift. The Norwegian Centre for Research Data approved the project. The data were gathered in three different ways: (1) the personal assistants were interviewed before and after one randomly chosen activity; (2) the activity was observed (consecutive reflection notes); and (3) the re-creation of and communication about this activity were recorded using three video cameras to capture (a) Magne's communication spheres of interest and (b) the assistant's communication. The semiformal interviews were based on an overall interview guide [30]. The interviewees were not interrupted when they spoke about their jobs and their interactions with Magne; therefore, the interviews may not have always complied rigidly with the overall guide, but the engagement shown when speaking freely might have value in adding to our understanding of the relations between Magne and the assistants.

### 4.3. Data Analysis

In the data analysis, the video recordings were viewed several times. These could not be transcribed because Magne uses neither verbal nor sign language, and his utterances in the form of frowns, eye focus, smiles, and others would disappear in a transcription. The

interviews were transcribed. The names Magne, Anna, and Bendik are anonymised, and the genders were changed in the text to ensure internal and external anonymity.

The next step in the process was to read and code the data in an open analysis [31]. This resulted in thirteen different codes and many comments. The data were reduced when the codes were thematised [32] and sorted in a process of axial coding [31] as we were searching for indications or contraindications of fellowship, dialogues, and communication as an activity. We were also open to other themes of significance for the research question. After some repetitions of data reduction, we ended up with the following categories: (a) feelings of confidence; (b) shared repertoire for communication; and (c) communication with meaning. These categories are continued into the structure of the presentations of the findings below.

## 5. Analysis, Results, and Discussion

As described above, the analysis is based on interpretations of the empirical data. The interpretation starts early in the process and gradually proceeds into a discussion of the results. Therefore, in this section, we show some of the analyses and present and discuss the results in each category. The structure of this section follows the categories uncovered in the analysis: (a) feelings of confidence; (b) shared repertoire for communication; and (c) communication with meaning.

### 5.1. Feeling Confident

Anna and Bendik, the two personal assistants interviewed, repeatedly expressed the importance of Magne feeling confident. Words such as 'confident' and 'avoiding insecurity' occurred eight times in 'Interview One with Anna' and 'Interview One with Bendik', where Anna and Bendik talked in general terms about their relationships and communication with Magne. If Magne is not confident, communication does not work.

> I feel that confidence is important . . . If I make a promise, I have to keep it . . . If not, he might feel unconfident and distrust me. I have never misled him. That would be stupid because then he would not trust me. Anna (int 1, p. 6)

Anna and Bendik both expressed the importance of relying on one another, and appropriate communication with Magne builds on this. Magne's engagement in the dialogues supports his sense of mastery and learning [10]. Based on the responses, the assistants know whether the communication makes meaning [9], and the communication shows some of the characteristics described by Alexander [15]. A lack of meaning and distrust from Magne seem to lead to non-acceptable acting out. Magne experiences rigid rules and regulations but also respects and, to a certain level, co-determination and participation [16]. Through staging [2], Anna and Bendik re-create the shared experience; in this way, they establish confidence in the situation, which is what Umagami [3] described as *recognition and safety* (p. 2129).

Belonging to a fellowship increases one's feeling of confidence, but at the same time, one needs confidence to first work for belonging, as Wenger [8] put it. Anna and Bendik talked about how Magne slowly built his confidence in the fellowship; agreements were made and kept, and the engagement increased. Magne started as an observer and moved slowly towards being a participant, correcting the personal assistants when they built the shared story about shared activities. As Wenger [8] stated, they were working for belonging based on confidence.

Confidence might be difficult to obtain when you lack a shared language. Bendik stated the following:

> I feel an ability to read his (Magne's) body language is necessary to facilitate and support confidence and understanding. Bendik (int 1, p. 1)

Confidence depends on understanding, which is based on body language in this case. When understood, Magne feels confident. If, in a learning process, Bendik can develop the use of symbols and other AAC tools, Magne's possibilities for negotiations and

dialogues would increase, and his influence on the situation and possibly his confidence would improve.

Magne is not part of an ordinary class. The confidence and trust between the personal assistants and Magne seem vital in establishing this compensating fellowship, and we need to regulate and adapt our expectations to Magne's contributions and need for confidence. Participations in the fellowship are far from equal, but framed by confidence, we might be able to expand this and increase equality. Anna and Bendik work to create a setting that elicits confidence for Magne. The difficulties that they encounter when communicating with Magne and having to resort to reading his body language are further reasons why they stress the importance of confidence in their interviews. They understand the need for a shared repertoire to create dialogic communication [15] with meaning [9] in an atmosphere of confidence. The process of re-creating a shared experience supports shared understanding. Again, this seems to support the feeling of confidence for all the participants.

Confidence seems important to experience belongingness in a fellowship. Communication and the experience of being understood are important for the possibility of participating in a fellowship.

*5.2. Shared Repertoire for Communication*

Wenger [8] presented a shared repertoire for communication as a characteristic of participants in a community of practice. The members should be able to negotiate meaning, interact, and tell stories about shared experiences. To do so, they need a form of shared language. We understand from the interviews that Anna and Bendik are also concerned about shared tools for communication.

Anna shared that Magne actively seeks the AAC boards now and that he has gained some understanding of what communication is:

> We talk about what has happened during the day ... He (Magne) laughs when we talk about events that have happened ... I understand that he understands what I am trying to tell him. He also seeks the boards quite actively now that we have worked with them for a while. In the beginning, I felt he was not too interested; he did not grasp the meanings. However, now, he initiates their use and tells us what he wants ... I really think we are on the right path. Anna (int 1, p. 1)

The assistants and Magne continuously develop their shared understanding and extend the use of symbols. They connect symbols with actions and concretes.

> We picked a stone, and then he (Magne) pointed to the stone symbol on the board. Bendik (int 2, p. 1)

On the walk, the objects and the symbols were connected. Magne participated in the communication when he was asked to point to the symbol [10,15]. Using shared symbols, a shared repertoire was developed [8]. In the reconstruction of the walk after returning home, stones were drawn, and the Playmobil figures were moved to the stones and 'thrown' into the water; photos were glued onto the drawings, and the AAC symbols were placed. Thus, a shared story and a shared language were built in a collaborative process [15]. Magne was engaged primarily as an observer, but he also corrected mistakes, thus participating in a reciprocal process [15]. In the communication and the reciprocal form, Magne experienced that communication is basically mutual. After years of not participating in dialogues, it was not initially obvious to him that communication is, in its simplest form, one person expressing something and the other person responding.

To benefit from what Wenger [8] postulated as the power of the community of practise, participants need a shared language. The development of a shared tool, a shared repertoire, and a language for communication as a fundament for understanding each other [14] was an important part of the special-needs education closely adapted to Magne's communicative competence. This is not just a specific need for Magne. Learning to communicate on a very basic level is important in the special education of other pupils in his situation. Because

these are not pupils who follow a common and well-known track of development, we, the teachers, personal assistants, parents, and researchers, have to work systematically to understand the learners' levels of communication and build on this with meaningful content in our communication for the pupils [9].

The constructional approach where the Playmobil figures were combined with drawing, with Anna and Bendik formulating a scene to communicate shared experiences, was a useful strategy for us; therefore, staging seems more widely applicable, but it has to be combined with the pupils' understanding of meaning [2]. The shared experiences frame a shared understanding of the content of the concepts. This development of a shared repertoire is possible in a fellowship with a shared focus and shared interests [8].

Shared language seems highly important to experience belongingness to a fellowship. The language has to be developed not only for the child but also for the personal staff. It is not enough to observe and use body language because the child's possibilities to express himself are severely reduced.

*5.3. Communication with Meaning and Utilitarian Value*

Through shared experiences, it became possible for Anne and Bendik to know more about what makes meaning to Magne. We concluded that if he does not respond at all to their communication, he either does not understand it or does not find the theme meaningful. When walking, it was extremely challenging to elicit a response except when picking up stones (Anne, int 2). Bendik also found a shared interest in throwing these stones.

> (When we) asked whether he wanted to throw stones into the water, he responded by doing it. (Bendik, int 2)

Magne did not communicate on the walk until the assistants picked up and threw stones into the water. The first response was to physically throw a stone. Later, when the action and communication were repeated, Magne communicated by pointing to the board, that is, he was 'speaking'. The communication was collaborative and reciprocal [15]. Bendik guided Magne to participate and build on their shared experience of throwing stones [19,20]. It looks like Magne depends on a very high level of personal utilitarian value to experience and understand the point of communication. Communicative actions must make meaning for him to be able to or want to participate [9,11].

We know from the situation described above by Anne and Bendik that Magne uses the boards actively in some situations to communicate about the 'here and now'. When it comes to communication about shared experience, the past and things that are not visible, it is more difficult for Magne to participate. Staging is a way to visualise and re-create the actual shared experience and communicate about it. When Magne, as shown in the video recordings, kept following with his eyes the person drawing, the process of placing the AAC symbols, and the photos on the large sheet of paper on the floor, we interpreted this as Magne being interested. He seemed to recognise the situation:

> First, he wanted to drive the car all over the drawings, but we talked about the parking space, and he parked the car and played [Bendik uses the verb 'to play' when we use the toy figures in our reconstruction. In a theoretical perspective, this is not playing but a play-like approach] with the figures. I am quite sure he connected the Playmobil figures, symbolising us as participants on the walk, especially when we cut out and glued small photos of our faces on each of the figures. We placed us (figures of the personal assistants) in the front seats, and he then placed the figure representing himself in the backseat. I am quite sure he associated the toys with the real objects: the car with his car, his figure as himself and so on. It is important to play (and communicate) like this: he recognises figures and moves around in the virtual world we constructed on the sheet of paper. Bendik (int 2, p. 2)

Anna and Bendik felt quite sure that Magne stayed interested because he understood the re-creation of the shared experience. He recognised the space that they created for

communication about things that happened earlier, which were no longer visible. This type of communication seemed to make meaning for Magne [6,11], although he did not obtain anything obvious from it, such as food. He engaged as part of the fellowship that went for a walk and communicated about it afterwards. The feeling of being part of a group, doing, and communicating about a shared activity were positive experiences for Anna and Bendik. Notably, it looked like it also affected Magne positively.

Anna and Bendik feel they are getting to know Magne better as they communicate more and share more experiences. When Magne and the personal assistants do things together and enhance their communication, they develop a shared repertoire and shared knowledge of the experiences represented by the symbols, photos, and drawings [18]. They share and develop a language, a repertoire for communication. Magne, Anna, and Bendik are engaged when communicating about these actions (mutual engagement). These two characteristics described by Wenger [8] appeared in the data analysis. The third characteristic, joint activity, was more difficult to reveal, but the walk and other events can be categorised as joint actions.

Categories embracing mutuality or collaboration were only observed to a minimal extent [8,15]. Magne was an observer more than a participator and collaborator in the communication.

> ( . . . ) he (Magne) continuously looked sideways and followed the drawings. If anything was wrong, he fixed it, for example, by moving the photos further on in the track. He followed the drawings like a hawk. However, today, he was not interested in participating in drawing, cutting or gluing. Bendik (int 2, p. 2)

When we repeated the exact same kind of activity with Magne, he seemed to recognise the form and gradually took a more active role. In this case, he regulated the visualising of the experience and corrected what appeared as mistakes to him. He participated in a dialogue where he responded adequately to utterings from the assistants [10]. Because the experience was shared, Anna (who was the person drawing) was able to respond to the corrections, disagree, or move the photos back. She could also ask, 'Why do you want to move this photo?' [12,13].

Magne did not move the communication and the actions forward but did make an almost imperceptible opening for further communication. This shows a possibility for communication based on shared experiences, but it also shows how demanding it is for Magne and the staff to establish dialogues. When working with communication as an educational goal, Anna and Bendik observed that Magne uses AAC when he has been trained and encouraged, and the response is useful and meaningful for him. The interest Magne shows in communicating about the shared experience can be interpreted as a sense of belonging to the fellowship.

## 6. Conclusions

These findings show that the reconstruction of a shared experience by drawing, painting, and moving Playmobil figures (i.e., staging) opens the door to participation in meaningful communication in a fellowship, even though the members are two adults and one boy with severe communicative disorders. The findings show that it is possible, even though it is not easy, to enter meaningful communicative settings and develop communicative skills and competence together. The staging enabled Magne to express himself a little more. The shared experiences scaffolded the confidence. The adults captured Magne's subtle utterings, and the shared experiences scaffolded the shared understanding and communication based on a shared repertoire of tools that supplied the communication with mutual engagement and meaning. Inappropriate communication, such as acting-out behaviours, decreased.

Magne communicates the results in his own way. Inappropriate acting-out behaviours decreased, and we interpret this as documentation for communicative development. When he uses some AAC symbols on his own initiative, this is also interpreted as a sign of increasing communicative skills. This way, Magne has shown us his experience of belonging to a fellowship. Children like Magne need a language and need to communicate. Otherwise,

they may be locked into a downward spiral of decreasing communication and increasing acting out and feel lonely and isolated. When belonging to a fellowship, quality of life seems to increase, and inappropriate actions decrease. An upward spiral is implemented and might help Magne become part of an even larger shared community.

We conclude that communication about a shared activity and shared experiences seems to lead to interest and engagement and, ultimately, a shared feeling of 'doing something together' and being part of a fellowship.

**Author Contributions:** Both authors have contributed equally during all stages in the process of writing except for the translation into English, this is done by the first author. All authors have read and agreed to the published version of the manuscript.

**Funding:** This research received no external funding.

**Institutional Review Board Statement:** The study was conducted in accordance with the Declaration of Helsinki and is approved by Norwegian statistical data (earlier NSD, now SIKT) with 449385 as the reference number.

**Informed Consent Statement:** Written and oral informed consent was obtained from Magnes parents and all his personal assistants to publish this paper.

**Data Availability Statement:** For ethical reasons, data is not public in this project. The data is stored on the encoded server (P:) of University of Southeast Norway.

**Conflicts of Interest:** As informed of in the text, both authors are related to Magne. No economical conflict of interest.

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
