# Peer review of "Staging as Communicative Activity of Shared Experiences a Way into a Fellowship for Deaf Children with Autism"

_education, doi:10.3390/educsci13040413_

Round 1

Reviewer 1 Report

The article analyzes a very relevant problem related to the development of scientific knowledge and the development of educational practice. However, I would like to make some suggestions to improve this article.

The structure of this article should meet the requirements of a scientific article. The problem analyzed in the article should first of all have a detailed theoretical justification, which should be strengthened in connected with the problem under consideration and presented before the methodology part. The research context should be be transferred after the theoretical part.

Author Response

Thank you for inspiring review. We have made major editions in the text. The main points are listed beneath.

  • The structure is changed, now the theoretical part is placed before the description of the context. The description of the context has been rearranged and is now concentrated in one section.
  • The research question is more precisely formulated.
  • The rearrangement led to changes within each section.
  • Theory has been added to justify the research and the research question.
  • The research method is more explicitly described, ethical perspectives are strengthened.
  • The text is proof read

Reviewer 2 Report

The paper adresses a relevant topic of autism, but it could be theoretically and methodologically more elaborated. The theoretical constructs of autism are not well enough precised. More actual literature could be integrated. The empirical study is not driven by theoretically sound questions but seems rather an evaluation of a case study.

Author Response

Thank you for your review. We have made major editions in the text. The main points are listed beneath.

  • The structure is changed, now the theoretical part is placed before the description of the context. The description of the context has been rearranged and is now concentrated in one section.
  • The research question is more precisely formulated.
  • The rearrangement led to changes within each section.
  • Theory has been added to justify the research and the research question.
  • The research method is more explicitly described, ethical perspectives are strengthened.
  • The text is proof read

Round 2

Reviewer 1 Report

Your manuscript meets the quality requirements.

Reviewer 2 Report

The paper has improved its scientific quality with the changes made.